# Overexpression of Toxic Poly(Glycine-Alanine) Aggregates in Primary Neuronal Cultures Induces Time-Dependent Autophagic and Synaptic Alterations but Subtle Activity Impairments

**DOI:** 10.3390/cells13151300

**Published:** 2024-08-03

**Authors:** Christina Steffke, Shreya Agarwal, Edor Kabashi, Alberto Catanese

**Affiliations:** 1Institute of Anatomy and Cell Biology, University of Ulm, 89069 Ulm, Germany; christina.steffke@uni-ulm.de (C.S.); shreya.agarwal@imtek.uni-freiburg.de (S.A.); 2Department of Neurology, University of Ulm, 89069 Ulm, Germany; 3International Graduate School in Molecular Medicine of Ulm (IGraDU), University of Ulm, 89069 Ulm, Germany; 4Institute Imagine, Necker-Enfants Malades Hospital, University Paris Descartes, 75015 Paris, France; edor.kabashi@institutimagine.org

**Keywords:** C9orf72, ALS, poly(GA), autophagy, synapse, MEA

## Abstract

The pathogenic expansion of the intronic GGGGCC hexanucleotide located in the non-coding region of the *C9orf72* gene represents the most frequent genetic cause of amyotrophic lateral sclerosis (ALS) and frontotemporal dementia (FTD). This mutation leads to the accumulation of toxic RNA foci and dipeptide repeats (DPRs), as well as reduced levels of the C9orf72 protein. Thus, both gain and loss of function are coexisting pathogenic aspects linked to *C9orf72*-ALS/FTD. Synaptic alterations have been largely described in *C9orf72* models, but it is still not clear which aspect of the pathology mostly contributes to these impairments. To address this question, we investigated the dynamic changes occurring over time at the synapse upon accumulation of poly(GA), the most abundant DPR. Overexpression of this toxic form induced a drastic loss of synaptic proteins in primary neuron cultures, anticipating autophagic defects. Surprisingly, the dramatic impairment characterizing the synaptic proteome was not fully matched by changes in network properties. In fact, high-density multi-electrode array analysis highlighted only minor reductions in the spike number and firing rate of poly(GA) neurons. Our data show that the toxic gain of function linked to *C9orf72* affects the synaptic proteome but exerts only minor effects on the network activity.

## 1. Introduction

C9orf72 is a protein ubiquitously expressed across the central nervous system, whose biological functions, despite its abundance, still remain puzzling. From a structural point of view, C9orf72 is characterized by a DENN domain [1], which, together with the interaction with its binding partners, SMCR8 and WDR41, indicates a role as a guanine nucleotide exchange factor (GEF). In fact, the C9orf72-SMCR8-WDR41 complex has been linked to TBK1-dependent autophagy through the activation of RAB39b [2], highlighting an important role played in the maturation of catabolic vesicles. Notably, Rab proteins are also involved in the regulation of synaptic vesicle (SV) dynamics, which are required to maintain proper neuronal activity. Since DENN domains are required for the regulation of Rab GTPases [3], the possible contribution of C9orf72, which has been indeed localized at the synapse, in the control of SVs is reasonable. Accordingly, it has been shown that C9orf72 interacts with synapsin and its deletion leads to reduced firing properties [4]. Thus, it appears that the structural organization of C9orf72 links this protein to at least two major biological processes involved in neuronal homeostasis: autophagy and synaptic activity. These are detrimentally affected in different neurodegenerative diseases including amyotrophic lateral sclerosis (ALS) and frontotemporal dementia (FTD), where mutations within the *C9orf72* gene represent the most frequent genetic cause [5]. Interestingly, the pathogenic mutations characterizing *C9orf72* occur in the intronic region located between the exons 1a and 1b of the gene. This non-coding region is characterized by the repetition of a GGGGCC sequence, which is abnormally expanded in patients suffering from ALS and FTD [6,7]. The consequences of this hexanucleotide expansion are diverse; on one side, the GGGGCC repeats can be transcribed to form toxic RNA inclusions (foci) or even translated into five different dipeptide repeats (DPRs; depending on the directionality and reading frame) without the need for a starting codon [8]. On the other side, the presence of the expanded repeats seems to alter the methylation profile of the *C9orf72* gene itself, resulting in haploinsufficiency [9,10]. This indicates that the pathological mechanisms behind *C9orf72* mutations include both gain and loss of function, but a clear dissection of the pathobiochemistry triggered by GGGGCC expansion is still missing. In addition, it is still unclear how and to which extent the molecular alterations observed in *C9orf72* models (and patients) relate to the toxic gain of function or to *C9orf72* haploinsufficiency. As a consequence, the development of effective therapeutic strategies against *C9orf72*-ALS/FTD has been extremely challenging and, until now, unsuccessful [11]. Thus, a better understating of the exact molecular alterations linked to the different pathological aspects associated with hexanucleotide expansion might improve the development of effective treatments. In this study, we aimed to dissect the time-dependent alterations occurring at the synapse in a gain-of-function model of C9orf72 and to relate them to the autophagic defects triggered by the presence of toxic aggregates. To this end, we overexpressed the most abundant DPR, poly(Glycine-Alanine) (poly(GA), in primary cortical neurons and investigated how these structures affect the synaptic proteome and activity over time.

## 2. Materials and Methods

### 2.1. Primary Rat Cortical Neurons 

Primary cultures of rat cortical neurons were obtained from rat embryos (Sprague Dawley rats, Janvier Laboratories, Le Genest-Saint-Isle, France) at embryonic day 19, as described previously [12]. Under stereomicroscopic guidance, the cerebral cortices from both hemispheres were manually dissected and collected in cold HBSS. Then, the tissues were incubated with 0.25% trypsin-EDTA (Gibco, Thermo Fisher Scientific, Waltham, MA USA) for 10 min at 37 °C, followed by washing three times with DMEM (Gibco, Thermo Fisher Scientific) containing 10% fetal bovine serum (Gibco, Thermo Fisher Scientific), 1% penicillin/streptomycin (Gibco, Thermo Fisher Scientific), and 1% L-glutamine (Gibco, Thermo Fisher Scientific) (henceforth DMEM+). Afterwards, the cortices were mechanically dissociated in DMEM+ by repeated pipetting until a homogenous resuspension was achieved. The resuspended cells were filtered through a 70 µm mesh filter and counted with an Improved Neubauer counting chamber. Finally, 20,000 cells/well in 24-well plates (for staining) or 80,000 cells/well in 12-well plates (for Western Blot) were seeded in DMEM+ on glass coverslips or plates previously coated with poly-L-lysine (Sigma-Aldrich Chemie GmbH, St. Louis, MO, USA) (incubation with PLL for one hour at 37 °C; then, three washing steps with double distilled water). After letting the cells attach for three to four hours, the DMEM+ was replaced by Neurobasal Medium (Gibco, Thermo Fisher Scientific), containing 1% penicillin/streptomycin, 1% L-glutamine, and 2% B27. Neuronal cultures were maintained at 37 °C in 5% carbon dioxide until the planned experiments were performed.

To selectively overexpress toxic protein aggregates in primary neurons, we used an AAV9 allowing for the expression of the transgene under the control of the human Synapsin 1 promoter. The transduction of neurons was performed at DIV10 with either AAV9-hSyn-poly(GA)175-EGFP or AAV9-hSyn-EGFP as control (a gift from Bryan Roth; Addgene viral prep # 50465-AAV9). The AAV9-hSyn-poly(GA)175-EGFP was produced by the Penn Vector Core (University of Pennsylvania, Philadelphia, PA, USA). The transduction efficiency of the AAV9-hSyn-poly(GA)175-EGFP vector was previously assessed at 78% [13].

### 2.2. Immunocytochemistry

Immunostainings were performed as previously described [13]. At the respective time point, neurons were fixed with 4% paraformaldehyde (containing 10% sucrose), followed by incubation with a blocking solution (PBS + 10% Goat Serum + 0.2% Triton-x100) for two hours. Subsequently, cells were incubated with primary antibodies diluted in the same blocking solution overnight (24 h at 4 °C). Following three washing steps with PBS, the cells were incubated with secondary antibodies (diluted 1:1000 in PBS) for two hours at room temperature. Again, cells were washed 3 times before mounting with ProLong^TM^Gold Antifade reagent with DAPI (Invitrogen, Thermo Fisher Scientific, P36935).

### 2.3. Microscopy

Confocal microscopy and image acquiring were performed with a Leica DMi8 laser-scanning microscope (Leica Microsystems, Wetzlar, Germany) equipped with an ACS APO 63 × oil DIC immersion objective and the Leica Application Suite X (LasX) software (https://www.leica-microsystems.com/de/produkte/mikroskop-software/p/leica-las-x-ls/, Leica Microsystems, Wetzlar, Germany). Images were captured with a resolution of 1024 × 1024 pixels and a number of z-stacks (step size of 0.3 μm) spanning the whole-cell soma.

### 2.4. Image Analysis

After collapsing the z-stack using the maximum intensity projection tool of ImageJ (Version 1.54g, National Institutes of Health and the Laboratory for Optical and Computational Instrumentation, Wisconsin, Madison, WI, USA), 3 different dendrites of each neuron were randomly selected. An ROI was defined along the dendrite and the number and intensity of the synaptic markers Bassoon and Homer1 were traced with the ImageJ plugin FindFoci, using the Max Entropy algorithm. Finally, the resulting data were normalized to a ROI length of 10 μm.

For the analysis of pictures belonging to the same experiment and for figure display performed with ImageJ, the computational parameters and post-acquisition modifications were set equally.

### 2.5. Western Blot

Protein concentration was determined using the Bradford protein assay. For separation, an equal amount of protein was loaded on 8, 10, or 12% acrylamide SDS-PAGE gels, which were then transferred to a nitrocellulose membrane using a Trans-Blot Turbo device (Bio-Rad Laboratories, Inc., California, CA, USA). Non-specific binding sites were blocked by incubation with a 5% BSA solution (diluted in TBS, pH 7.5, +0.2% TWEEN) for two hours; then, membranes were incubated with the primary antibody overnight at 4 °C. Subsequently, blots were washed 3 times with TBS + 0.2% TWEEN, incubated with HRP-conjugated secondary antibody (DAKO, Glostrup, Denmark) for 1 h, and washed for another 3 times. Chemiluminescent signal was detected with a MicroChemi 4.2 device (DNR Bio Imaging System, Neve Yamin, Israel) using an ECL detection kit (ThermoFischer Scientific, 32106). For quantification, Gel-analyzer Software 19.1 was used. The levels of the proteins of interest were normalized against the loading control β-actin.

### 2.6. Antibody List 

In this study, the following primary antibodies were used: anti-mTor (diluted 1:500; Abcam, Cambridge, UK, ab32028), anti-p-mTor (diluted 1:500; phospho S2448; Abcam, ab109268), anti-p62 (diluted 1:1000; Abcam, Ab-56416), anti-Beclin (diluted 1:1000; Novusbio, Littleton, CO, USA, NB500-249), anti-Atg5 (diluted 1:500; Abcam, ab108327), anti-C9orf72 (diluted 1:1000; Genetex, Irvine, CA, USA GTX632041), anti-PSD95 (diluted 1:1000; Abcam, ab2723), anti-Gephyrin (diluted 1:1000; Synaptic Systems, Goettingen, Germany, 147 011), anti-Synapsin1-2 (diluted 1:500; Synaptic Systems, 106003), anti-SNAP47 (1:500; Synaptic Systems, 111403), anti-Synaptophysin (diluted 1:1000; Abcam, ab14692), anti-GluA1 (diluted 1:500; Synaptic Systems, 182003), anti-GluA2 (diluted 1:500; Synaptic Systems, 182211), anti-MAP2 (diluted 1:3000; Encor, Gainesville, FL, USA, CPCA-MAP2), Homer1 b/c (diluted 1:500; Synaptic Systems, 160 025), and anti-Basoon (diluted 1:1000; Enzo Life Sciences, New York, NY, USA, SAP7F407).

For Western blot experiments, HRP-conjugated anti-Mouse or anti-Rabbit purchased from DAKO were used.

For immunostainings, the following secondary antibodies from ThermoFisher Scientific were used at a dilution of 1:1000 if not otherwise mentioned: goat anti-Chicken DyLight 350 (SA5-10069), goat anti-Mouse Alexa Fluor^®^ 488 (A-11001), goat anti-Rabbit Alexa Fluor^®^ 488 (A-11008), goat anti-Mouse Alexa Fluor^®^ 568 (A-11004), goat anti-Rabbit Alexa Fluor^®^ 568 (A-11011), goat anti-Chicken Alexa Fluor^®^ 647 (diluted 1:50, A32933), goat anti-Mouse Alexa Fluor^®^ 647 (A-21235), and goat anti-Rabbit Alexa Fluor^®^ 647 (A-21244).

### 2.7. Multielectrode Array

Prior to plating, MaxWell MaxTwo 6-well plates were treated with sterile filtered Tergazyme (Alconox, White Plains, NY, USA) 1% solution at 37 °C overnight in order to increase the hydrophilicity of the surface. After removal of the Tergazyme solution and disinfection with 70% ethanol for 15–20 min, the chips were rinsed three times with PBS (−/−) before coating each active sensing area with a 50 µL drop of Poly-L-Lysine for 2 h at 37 °C. Afterwards, the coating solution was removed, and 3 washes with sterile water were performed. Neurons were plated at a density of 100.000 cells/wells by carefully seeding a drop of DMEM with the cells only on the electrode area. Neurons were then incubated for 2 h at 37 degrees before adding 500 µL of NB+. One day after plating, 1000 µL of NB+ were added to each well, and, starting from DIV15, a 25% medium change was conducted once a week. The presented data originate from a minimum of three independent cultures. Electrophysiological parameters were obtained using a MaxTwo Multiwell HD-MEA system (MaxWell Biosystems AG, Zurich, Switzerland). The system’s gain was set to 512× with a high-pass filter from 300 Hz and a spike threshold of 5.00. During recordings, a temperature of 37 °C was maintained. Starting from DIV14, the activity of the plates was monitored with weekly full-sensor “Activity Scan” assays in MaxLab Live software (https://www.mxwbio.com/products/maxlab-live-software/) with a record time of 20 s per electrode. If a well presented less than 2.5% active electrodes, it was not considered for further analysis. After the activity scan, we performed “Network Scan” assays (in MaxLab Live software) at the respective time point. Data shown in this paper were exclusively obtained from the “Network Scan” assays, where only the most active subset of electrodes per chip (as evaluated by a built-in Algorithm in MaxLab Live software, based on the firing rate) was recorded over a period of 300 s. At DIV28, AxonTracking Assay was performed on the basis of a full-sensor Activity Scan.

### 2.8. Data and Statistical Analysis

Data collection and statistical analysis were performed using Microsoft Excel (Version 16.80, Redmond, WA, USA), MaxLab, and GraphPad Prism (Version 8, Boston, MA, USA)).

All experiments were performed in a minimum of *N* = 3 independent replicates (independent preparations of primary cells), and the following statistical tests were applied: two independent groups were compared using the unpaired *t*-test with Welch correction in cases of normally distributed data and a nonparametric Mann–Whitney test in cases of non-normal distribution. Two-way ANOVA followed by the Holm–Šídák test for multiple comparisons was applied when two independent experimental groups were analyzed at different time points. Statistical significance was set at *p* < 0.05.

## 3. Results

### 3.1. Poly(GA) Overexpression Induces Time-Dependent Autophagic Alterations Independent from C9orf72 Levels

To investigate the time-dependent effects linked to the toxic gain of function associated with C9orf72, we used an AAV9 vector to overexpress poly(GA) aggregates (or EGFP as control—henceforth, GFP) with high efficiency in primary cortical neurons [13]. Neuronal cultures were transduced at day in vitro (DIV) 10, and a progressive increase in the expression of the transgene was detectable starting from DIV 14. Thus, we selected this as the first time point for our investigations and traced the consequences of poly(GA) accumulation also at DIV 21 and 28 (Figure 1A). Poly(GA) aggregates are known to sequestrate the autophagy receptor SQSTM1/p62, and, indeed, we could observe higher levels (close to statistical significance) of this protein in poly(GA) than in GFP-expressing cultures at DIV 14 (DIV14-PolyGA 1.677 fold of DIV14-GFP, Figure 1B,C). In accordance with the progressive accumulation of the aggregates, SQSTM1 displayed a dramatic increase over time in the poly(GA) neurons, confirming the interplay between the toxic DPR and the autophagy receptor (DIV21-PolyGA 3.626 fold of DIV21-GFP, *p* = 0.0010; DIV28-PolyGA 2.976 fold of DIV28-GFP, *p* = 0.0070). We then investigated the dynamics of other two autophagy proteins involved in the early stages of the flux: Beclin and Atg5. Interestingly, the levels of both proteins were comparable in GFP and poly(GA) cultures at DIV 14 and displayed a time-dependent reduction in both groups, which became significantly larger upon poly(GA) overexpression at DIV 28 (Beclin: DIV28-PolyGA 0.340-fold of DIV28-GFP, *p* < 0.0001; Atg5: DIV28-PolyGA 0.398-fold of DIV28-GFP, *p* = 0.0039). This suggested that cortical neurons display a reduction in the autophagic flux over time, which is detrimentally worsened by the presence of toxic aggregates. Interestingly, and in agreement with the reduced levels of autophagy suggested by Beclin and Atg5, the levels of phospho-mTor dynamically increased from DIV 14 to 28 in GFP cultures but not in poly(GA) ones (DIV28-PolyGA 0.272 fold of DIV28-GFP, *p* = 0.0057). Since total mTor displayed minor variability and remained comparable between the groups across the time points investigated, the failed activation of mTor triggered by poly(GA) indicated an overall catabolic blockade when aggregates accumulated in neurons over two weeks. Notably, this pathologic phenotype appeared to be independent from the endogenous C9orf72, whose levels were comparable between GFP and poly(GA) cultures at all the time points (Figure 1D).

### 3.2. Loss of Synaptic Proteins Anticipates Autophagic Defects in Poly(GA)-Expressing Cultures

Since synaptic defects have been widely observed in different disease models related to C9orf72, we investigated with which timing the accumulation of poly(GA) affects this microanatomical structure. To this end, we followed the levels of the presynaptic markers synaptophysin (Syp), synapsin 1-2 (Syn1-2) and synaptosome-associated protein 47 (Snap47), as well as the post synaptic proteins PSD-95, Gephyrin, and Homer1 at DIV 14, 21, and 28 in GFP and poly(GA)-expressing cultures (Figure 2A,B). Interestingly, the accumulation of the toxic aggregates led to a significant loss of all the synaptic proteins investigated already at DIV21, thus anticipating most of the autophagy-related impairments observed upon the same experimental conditions (Syp: DIV21-PolyGA 0.348 fold of DIV21-GFP, *p* = 0.0099; Syn1: DIV21-PolyGA 0.499 fold of DIV21-GFP; Syn2: DIV21-PolyGA 0.247 fold of DIV21-GFP; Snap47: DIV21-PolyGA 0.337 fold of DIV21-GFP, *p* = 0.0740; PSD-95: DIV21-PolyGA 0.342 fold of DIV21-GFP; Gephyrin: DIV21-PolyGA 0.032 fold of DIV21-GFP; and Homer1: DIV21-PolyGA 0.372 fold of DIV21-GFP, *p* = 0.0640). Moreover, the difference in the synaptic proteome of GFP and poly(GA) cultures became even more dramatic at DIV 28 (Syp: DIV28-PolyGA 0.105 fold of DIV28-GFP, *p* = 0.0409; Syn1: DIV28-PolyGA 0.151 fold of DIV28-GFP, *p* = 0.0493; Syn2: DIV28-PolyGA 0.059 fold of DIV28-GFP, *p* = 0.0534; Snap47: DIV28-PolyGA 0.038 fold of DIV28-GFP, *p* = 0.0360; PSD-95: DIV28-PolyGA 0.065 fold of DIV28-GFP, *p* = 0.0288; Gephyrin: DIV28-PolyGA 0.020 fold of DIV28-GFP, *p* = 0.0460; and Homer1: DIV28-PolyGA 0.136 fold of DIV28-GFP, *p* = 0.0268; Figure 2C). In fact, while the synaptic proteins became more abundant in GFP-expressing neurons over time, the presence of poly(GA) structures induced a progressive loss of the same markers in the culture’s total lysates (DIV28-PolyGA 0.020 fold of DIV28-GFP, *p* < 0.0001; Figure 2D), as well along the dendrites of single neurons (Basoon: PolyGA 0.64 fold of GFP, *p* < 0.0001; Homer1: PolyGA 0.56 fold of GFP, *p* < 0.0001; Appendix A).

### 3.3. The Network Properties of Primary Cultures Are Minimally Affected by Poly(GA) Overexpression

Based on the drastic reduction in synaptic proteins observed upon poly(GA) overexpression, we speculated that the electrophysiological properties of these cultures might be largely affected as well. To answer this question, we used a multi electrode array (MEA) MaxTwo system (MaxWell Biosystem) to longitudinally monitor the activity of GFP and poly(GA) cultures at DIV 14, 21, and 28. At all these time points, the activity of primary neurons cultured onto MaxTwo 6-Well plates were first checked performing a full array-based activity scan. After this initial analysis, the most active electrodes (from the total 26.400 contained in the sensing area) were selected to perform a Network Analysis (Figure 3A), with which we analyzed activity parameters such as the firing rate, burst frequency, interspike interval (ISI) within burst and burst peak firing rate. In line with neuronal maturation, both GFP and poly(GA) cultures displayed a progressive increase in activity properties such as firing rate and burst frequency. Accordingly, the ISI within burst decreased over time, indicating increased activity, while the burst peak firing rate was stable across the time points (Figure 3B). Surprisingly, all these parameters remained comparable between GFP and poly(GA) cultures at all time points, which indicated general electrophysiological stability in the neuronal networks despite the presence of toxic aggregates. These unexpected results appeared to be in strong contrast with the loss of synaptic proteins observed in the total lysate of poly(GA) cultures. For this reason, we performed an Axon Tracking Assay with the HD-MEA to simultaneously track the action potential (AP) propagation in single neurons within DIV 28 GFP and poly(GA) cultures (Figure 3C). Even though this approach did not highlight any major morphological difference in the axonal parameters of both experimental groups (Appendix A), we could detect a significant reduction in the firing rate and number of spikes in poly(GA) neurons, which also displayed a significantly higher amplitude at the initiation site of AP propagation (Firing rate: PolyGA 0.7940 fold of GFP, *p* = 0.0027; number of spikes: PolyGA 0.8182 fold of GFP, *p* = 0.0027; amplitude: PolyGA 1.1722 fold of GFP, *p* = 0.0002; Figure 3D). This indicated that electrophysiological alterations occur only at later stages of poly(GA) accumulation, suggesting the presence of synaptic molecules that might be more resistant to the toxic effects triggered by the accumulation of aggregates. To prove this theory, we monitored the levels of the two AMPA-receptor subunits, GluA1 and GluA2, over time in GFP and poly(GA) cultures. In contrast to the scaffold and vesicle-related synaptic proteins (Figure 2), the levels of the GluA1 and GluA2 remained comparable in both experimental groups until DIV 21 and became drastically lower in poly(GA) than in GFP-expressing neurons only at DIV 28 (GluA1: DIV28-PolyGA 0.064 fold of DIV28-GFP, *p* = 0.032; GluA2: DIV28-PolyGA 0.497 fold of DIV28-GFP, *p* = 0.0485; Figure 4).

All in all, our results highlight that the synaptic alterations triggered by poly(GA) accumulation anticipate autophagic defects and affect the abundance of synaptic proteins involved in structure, vesicle dynamics, and, to a lesser and slower extent, glutamate receptors. As a consequence, the neuronal networks established in poly(GA) cultures display only minor signs of altered activity in comparison to neurons without aggregates.

## 4. Discussion

The recent failure of the antisense oligonucleotide-based trials against C9orf72 mutations, which represent the major genetic cause of ALS and FTD, has highlighted the need for a better understanding of the pathological features linked to this gene. In fact, it is even still debated whether pathogenic GGGGCC expansion actually leads to reduced levels of the C9orf72 protein, as haploinsufficiency has been inconsistently described [14]. This means that the complexity and interconnection of the molecular mechanisms triggered by hexanucleotide expansion must be better detangled in order to optimize therapeutic strategies aiming to correct the C9orf72-linked pathology. On the one hand, the toxicity of dipeptide repeats and RNA foci, which are directly linked to the GGGGCC repeats, has already been widely described and accepted as a core component of C9orf72 pathobiochemistry [15,16]. On the other hand, the loss of C9orf72 physiological functions seems to trigger additional alterations that also contribute to neuronal sufferance. The roles played by this protein in autophagy and synaptic dynamics [2,17,18], for example, indicate that catabolic and electrophysiological phenotypes might be directly linked to pathogenic mutation and do not represent the simple consequence of neurodegeneration. Considering that the exact function(s) of C9orf72 in neurons must still be fully understood, the dissection of the pathomechanisms linked to this gene has resulted in complex and often contradictory outcomes. In the case of catabolic impairment, reduced levels of C9orf72 lead to an early-phase blockade of the canonical autophagic pathway through the disruption of the WDR1-SMCR8 complex [2,19]. Still, Beckers and colleagues did not identify signs of autophagy impairment in C9orf72 knockout human motor neurons [20] and suggested a major contribution from the toxic gain of function to this phenotype. This appears to be at least partially in contrast with previous observations that linked C9orf72 haploinsufficiency to impaired endolysosomal degradation, which led to increased levels of NMDA synaptic receptors and vulnerability to glutamate excitotoxicity [21]. Despite these results possibly appearing in contrast at a first glance, they actually highlight how the interplay of both loss- and gain-of-function aspects of C9orf72 pathology synergistically contribute to neuronal vulnerability. In fact, restoring C9orf72 levels while boosting catabolism proved neuroprotective using in vitro models linked to this protein [22]. Nevertheless, it is still reasonable to consider that some alterations observed in mutant neurons might be stronger linked to the one or the other aspect of C9orf72 pathology. For example, its interaction with synapsin suggests a direct role played in the maintenance of functional synaptic contacts [4,18], also through the regulation of autophagy [23]. In this context, the results here presented contribute to a better understanding of the dynamic alterations affecting the synapse in C9orf72 disease models. The overexpression of toxic poly(GA) aggregates indeed affected the synaptic proteome before altering the activation of mTOR and the levels of Atg5 and Beclin, which are involved in the early stages of the flux [24]. Thus, besides the immediate sequestration of the autophagic receptor SQSTM1/p62, it seems that altered synaptic structure (as shown by the altered scaffolds) and vesicle-release machinery represent a response to the accumulation of toxic aggregates that anticipates major blockades in the macroautophagic pathway. Notably, the early loss of synaptic proteins detected in poly(GA) cultures was not timely matched by alterations of network electrophysiological properties, which were maintained until the later stages of degeneration when catabolic disruption is also observed. Interestingly, this seems to be mediated by the maintenance of physiological levels of (at least some) glutamate receptors, which dropped below the levels of GFP controls only at DIV 28. A possible explanation for this divergent loss of different synaptic proteins might be directly linked to the catabolic impairments linked to C9orf72 itself. In fact, reduced levels of C9of72 affect the efficacy of endolysosome-mediated degradation of glutamate receptors [21], and the acute overexpression of aggregates did not impact the levels of C9orf72 in poly(GA) cultures. This might reflect the selective degradation of specific synaptic substrates that directly depend on the levels of C9orf72. While its interaction with Rab5 [25] seems to be mostly required for endosomal function at the post-synapse, binding with the synapsin at the presynaptic site seems to be required for proper neuronal activity [4]. Thus, C9orf72 is involved in protein and vesicle turnover at both sites of the synapse, with a major catabolic shift at the postsynaptic side. In this speculative model, the accumulation of protein aggregates might first impact the presynaptic structure, which is commonly altered across the ALS spectrum [26], without affecting the levels of glutamate receptors whose physiological turnover is ensured by the presence of C9orf72. This appears to be sufficient to temporarily compensate for synaptic disfunctions, which become evident only once the overload of protein aggregates becomes intolerable for poly(GA) cultures and major autophagic impairments occur. In conclusion, the presence of toxic aggregates is sufficient to trigger a dramatic change in the synaptic proteome of affected neurons but has only a secondary impact on the electrophysiological properties of their network when C9orf72 is not altered. Nevertheless, it must be highlighted that these data were obtained by artificially overexpressing a toxic construct and that the use of more physiological models with an increased sample size might help in better dissecting the exact impact of DPRs to a synaptic pathology. Nevertheless, these results strengthen the idea that a synergistic coexistence of both loss- and gain-of-function mechanisms is required to fully trigger the pathobiochemical features linked to C9orf72 mutations.

## Figures and Tables

**Figure 1 cells-13-01300-f001:**
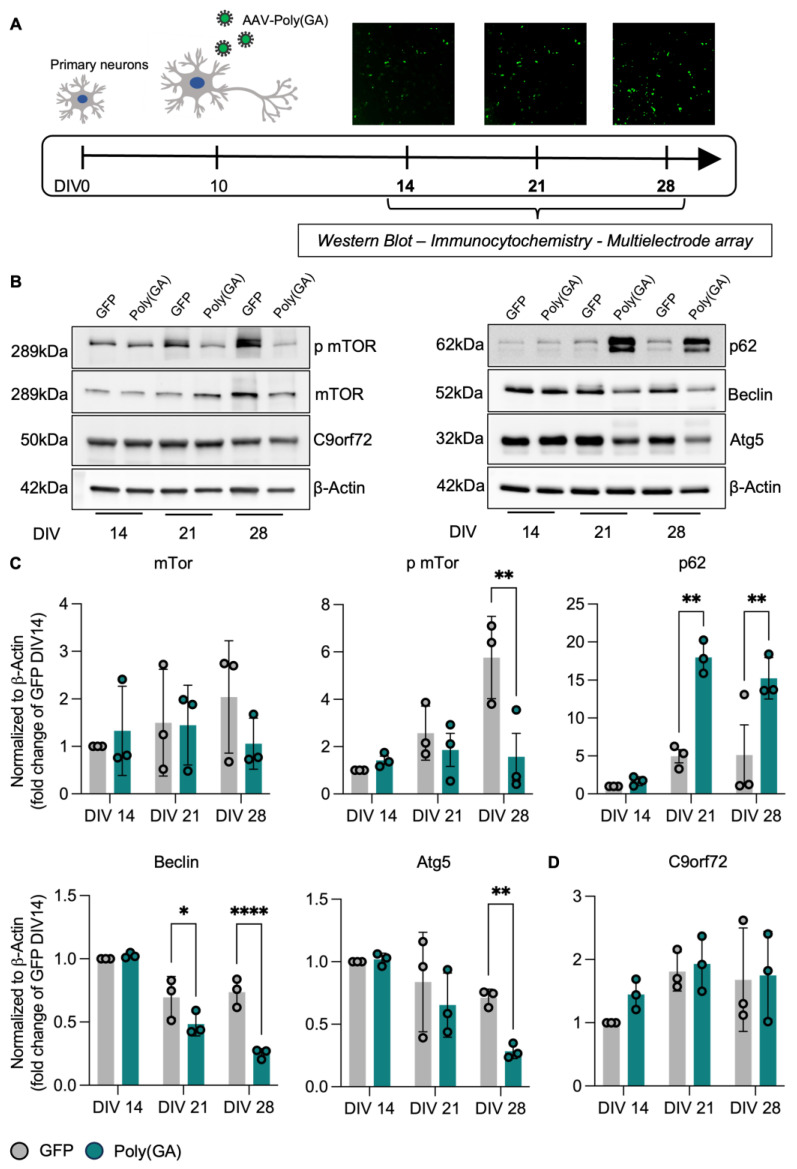
Time-dependent alterations in autophagic pathway upon poly(GA) overexpression independent from C9orf72 levels. (**A**) Workflow: Primary cortical neurons from rat embryos (E19) were plated at DIV0 and transduced with AAV9-poly(GA) to overexpress poly(GA) aggregates or AAV9-GFP as control at DIV10. Cells were cultured up to DIV 14, 21, and 28, and Western Blotting, immunocytochemistry, and microelectrode array (MEA) were performed at all 3 time points to elucidate changes over time. Microscope pictures were acquired with a 10x objective and an additional 3x digital zoom. (**B**) Representative blots. (**C**) Canonical proteins in early phases of autophagy, Beclin and Atg5, reveal a time-dependent decrease in poly(GA) and control cultures, with significantly lower protein levels at DIV28 upon toxic Poly(GA) aggregates (Beclin: DIV21-PolyGA vs. DIV21-GFP *p* = 0.0448; DIV28-PolyGA vs. DIV28-GFP *p* < 0.0001; Atg5: DIV28-PolyGA vs. DIV28-GFP *p* = 0.0039). Together with distinctly accumulating p62 protein over time (DIV21-PolyGA vs. DIV21-GFP *p* = 0.0010; DIV28-PolyGA vs. DIV28-GFP *p* = 0.0070), this indicates an impaired autophagic flux aggravated by poly(GA). Interestingly, cultures overexpressing poly(GA) lack increasing phosphorylated mTor within 2 weeks (DIV28-PolyGA 0.272 vs. DIV28-GFP *p* = 0.0057), while total mTor only shows minor differences, implying an overall catabolic blockade. Intriguingly, (**D**) C9orf72 levels remain comparable between both conditions. Protein levels were normalized to β-Actin levels as loading control. Values are shown as fold change to GFP DIV14. Experiments were performed in *N* = 3 independent replicates. Data are displayed as mean value ± SD (two-way ANOVA; * *p* < 0.05; ** *p* < 0.01; **** *p* < 0.0001; all other comparisons are not significant).

**Figure 2 cells-13-01300-f002:**
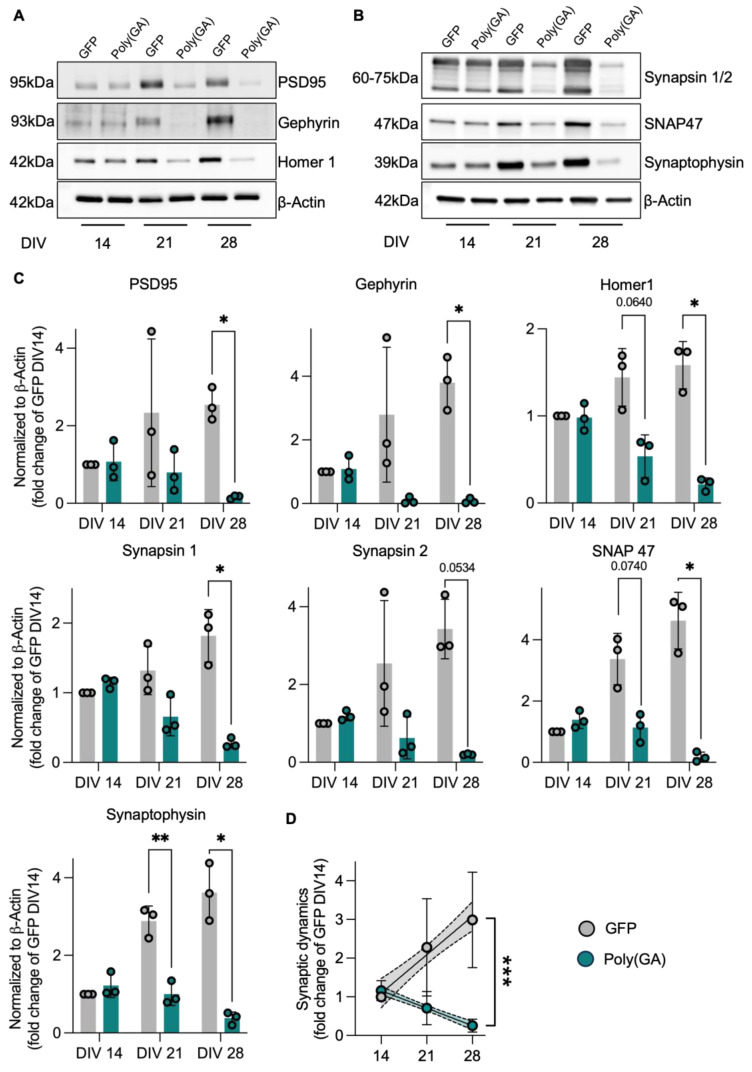
Poly(GA) aggregates induce reduction in pre- and postsynaptic proteins. Representative blots of (**A**) post- and (**B**) presynaptic proteins. (**C**) Protein levels of pre- and postsynaptic scaffold- and vesicle-related proteins show a significant decrease in poly(GA)-transduced cultures already at DIV21 (Syp: DIV21-PolyGA vs. DIV21-GFP *p* = 0.0099; Snap47: DIV21-PolyGA vs. DIV21-GFP *p* = 0.0740; Homer1: DIV21-PolyGA vs. DIV21-GFP *p* = 0.0640) with aggravation up to DIV28 (Syp: DIV28-PolyGA vs. DIV28-GFP *p* = 0.0409; Syn1: DIV28-PolyGA vs. DIV28-GFP *p* = 0.0493; Syn2: DIV28-PolyGA vs. DIV28-GFP *p* = 0.0534; Snap47: DIV28-PolyGA vs. DIV28-GFP *p* = 0.0360; PSD-95: DIV28-PolyGA vs. DIV28-GFP *p* = 0.0288; Gephyrin: DIV28-PolyGA vs. DIV28-GFP *p* = 0.0460; and Homer1: DIV28-PolyGA vs. DIV28-GFP *p* = 0.0268). The opposite synaptic dynamics of decreasing protein expression upon poly(GA), in contrast to increasing synaptic proteins in the sense of neuronal maturation over time in control cultures (DIV28-PolyGA vs. DIV28-GFP *p* < 0.0001), is highlighted in (**D**). Protein levels were normalized to β-Actin levels as loading control. Values are shown as fold change to GFP DIV14. Experiments were performed in *N* = 3 independent replicates. Data are displayed as mean value ± SD (2way-ANOVA; * *p* < 0.05; ** *p* < 0.01; *** *p* < 0.001; all other comparisons are not significant).

**Figure 3 cells-13-01300-f003:**
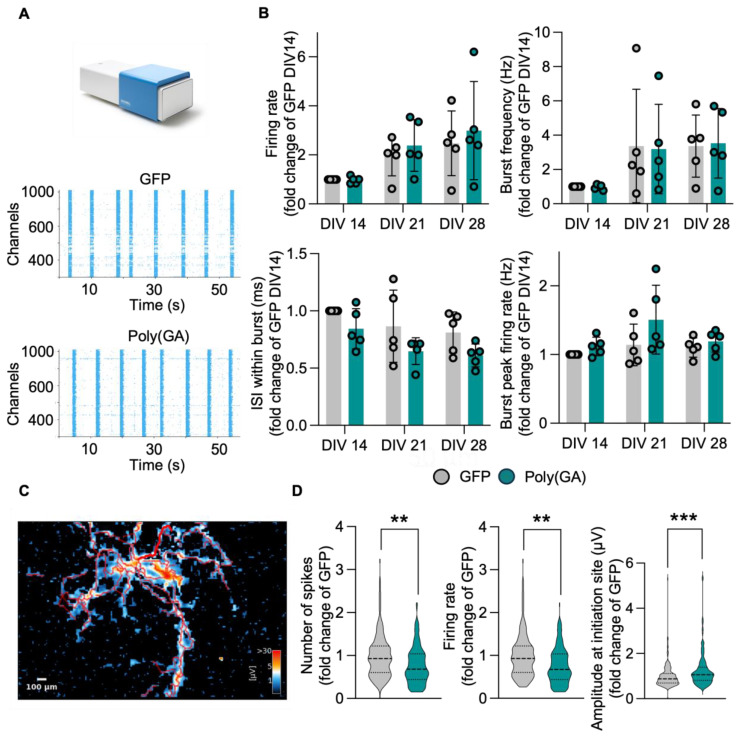
Subtle impairment of network properties in primary neurons upon poly(GA) exposition. (**A**) Representative network activity plots of neuronal cultures. (**B**) Surprisingly, network activity represented by firing property, interspike interval (ISI), burst frequency, and burst peak firing rate remains unaltered between control and poly(GA)-overexpressing cultures at all timepoints and even displays a maturing-dependent signature of increased activity. (**C**) Axonal tracking assay reveals (**D**) a significantly decreased number of spikes (PolyGA vs. GFP *p* = 0.0027) and firing rate (PolyGA vs. GFP *p* = 0.0027) at the axonal initiation site, while the amplitude is significantly higher (PolyGA vs. GFP *p* = 0.0002) in poly(GA)-transduced cultures. Values are shown as fold change to GFP (DIV14). Experiments were performed in *N* = 3 independent replicates. Data are displayed as mean value ± SD (two-way ANOVA or Welch’s *t*-test; ** *p* < 0.01; *** *p* < 0.001; all other comparisons are not significant).

**Figure 4 cells-13-01300-f004:**
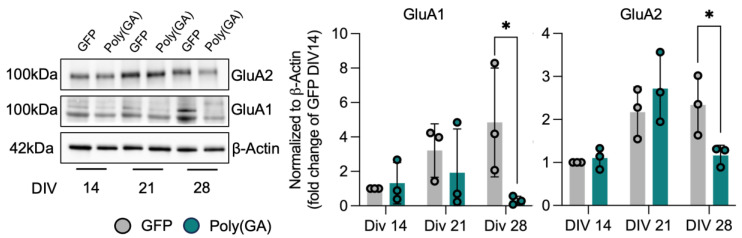
AMPA-receptor subunits seem to be more resistant to toxicity of poly(GA) aggregates. Protein levels of GluA1 and GluA2 display significant downregulation upon poly(GA)-transduction only at DIV28 (GluA1: DIV28-PolyGA vs. DIV28-GFP *p* = 0.032; GluA2: DIV28-PolyGA vs. DIV28-GFP *p* = 0.0485), while up until DIV21, the receptor subunits still show comparable levels between poly(GA) and GFP, in contrast to scaffold- and vesicle-related synaptic proteins. Protein levels were normalized to β-Actin levels as loading control. Values are shown as fold change to GFP DIV14. Experiments were performed in *N* = 3 independent replicates. Data are displayed as mean value ± SD (two-way-ANOVA; * *p* < 0.05; all other comparisons are not significant).

## Data Availability

All the data are included in the manuscript.

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
