# Peer review of "Overexpression of Toxic Poly(Glycine-Alanine) Aggregates in Primary Neuronal Cultures Induces Time-Dependent Autophagic and Synaptic Alterations but Subtle Activity Impairments"

_cells, 2024, doi:10.3390/cells13151300_

Round 1

Reviewer 1 Report

Comments and Suggestions for Authors

Ref: cells-3115693

Title: Overexpression of toxic poly(GA) aggregates in primary neuronal cultures induces time-dependent autophagic and synaptic alterations but subtle activity impairments

The study investigates the pathogenic expansion of the GGGGCC hexanucleotide in the C9orf72 gene, a leading genetic cause of amyotrophic lateral sclerosis (ALS) and frontotemporal dementia (FTD). This mutation causes the accumulation of toxic RNA foci and dipeptide repeats (DPR), particularly poly(GA), while also reducing C9orf72 protein levels. The research highlights both gain- and loss-of-function mechanisms in C9orf72-related ALS/FTD. Synaptic alterations were observed in C9orf72 models, with over-expression of poly(GA) leading to a significant loss of synaptic proteins and early autophagic defects in primary neuron cultures. Despite these severe synaptic protein impairments, network activity showed only minor reductions in spike number and firing rate. This study underscores the complex relationship between synaptic protein loss and network activity changes in C9orf72-linked neurodegeneration.

Comments:

Major comments:

1.      The results section lacks specific numerical data such as numbers or percentages. Each paragraph consists of general statements describing the results. It is important to highlight statistically significant changes.

2.      The results are obtained based on n=3 (mainly). The findings of the study should be interpreted with caution due to the small sample size.

Minor comments:

1.  Please provide detailed information on primary rat cortical neurons so that the experiments can be replicated by another group.

2.  Please add the dilutions of antibodies.

3.  Please provide the information about the effectiveness of transduction.

Author Response

The study investigates the pathogenic expansion of the GGGGCC hexanucleotide in the C9orf72 gene, a leading genetic cause of amyotrophic lateral sclerosis (ALS) and frontotemporal dementia (FTD). This mutation causes the accumulation of toxic RNA foci and dipeptide repeats (DPR), particularly poly(GA), while also reducing C9orf72 protein levels. The research highlights both gain- and loss-of-function mechanisms in C9orf72-related ALS/FTD. Synaptic alterations were observed in C9orf72 models, with over-expression of poly(GA) leading to a significant loss of synaptic proteins and early autophagic defects in primary neuron cultures. Despite these severe synaptic protein impairments, network activity showed only minor reductions in spike number and firing rate. This study underscores the complex relationship between synaptic protein loss and network activity changes in C9orf72-linked neurodegeneration.

Comments:

Major comments:

The results section lacks specific numerical data such as numbers or percentages. Each paragraph consists of general statements describing the results. It is important to highlight statistically significant changes.

Reply: we have now included in the results section and figure legends the fold change and exact p value for each significant change

2.      The results are obtained based on n=3 (mainly). The findings of the study should be interpreted with caution due to the small sample size.

Reply: we have added a sentence at the end of the discussion highlighting the limitation of our study

Minor comments:

Please provide detailed information on primary rat cortical neurons so that the experiments can be replicated by another group.

Reply: the details on primary neurons have been added as requested

Please add the dilutions of antibodies.

Reply: the dilutions have been added

Please provide the information about the effectiveness of transduction.

Reply: the transduction efficacy was assessed at 78% in our previous work (PMID: 34125498). This information is now also reported in the paragraph on primary neurons within the Methods section

Reviewer 2 Report

Comments and Suggestions for Authors

The authors research the important topic concerning the influence of poly(GA) buildup as a consequence of a pathogenic expansion of the intronic GGGGCC hexanucleotide in neurons associated with ALS and FTD.

The manuscript is well written and the results sound correctly presented.

I have no major comments on the manuscript, however I am not an expert in the methodology used.

Author Response

The authors research the important topic concerning the influence of poly(GA) buildup as a consequence of a pathogenic expansion of the intronic GGGGCC hexanucleotide in neurons associated with ALS and FTD.

The manuscript is well written and the results sound correctly presented.

I have no major comments on the manuscript, however I am not an expert in the methodology used

Reply: thanks for the positive feedback

Round 2

Reviewer 1 Report

Comments and Suggestions for Authors

The topic of the manuscript is both contemporary and highly relevant to current research in cell biology. The authors have addressed and revised their text according to previous feedback, significantly improving its quality.

I am pleased to recommend the publication for consideration in the journal Cells. Given the modern and significant nature of the subject matter, I believe that this manuscript would make a valuable contribution to the journal.